# Rare diseases in children—Knowledge, experiences and challenges faced by pediatricians in Tanzania

Mariam Noorani[1,2]*, Mohamedraza Ebrahim[1], Francis Furia[2]

**1** Department of Pediatrics, Aga Khan University, Dar es Salaam, Tanzania, **2** Department of Pediatrics and Child Health, Muhimbili University of Health and Allied Sciences, Dar es Salaam, Tanzania

\* mariam.noorani@aku.edu

## Abstract

Diagnosing and treating rare diseases in children is a major challenge for pediatricians globally. There is a lack of adequate knowledge of these conditions and diagnostic testing is not easily accessible, which frequently results in delays in care. The knowledge, experiences and challenges faced by pediatricians in Tanzania are not known. This study used a nationwide cross-sectional online survey to describe the knowledge of pediatricians in Tanzania on rare diseases, their experiences, and the challenges they face in treating these children. The survey tool was shared on the Pediatric Association of Tanzania WhatsApp group where most pediatricians are registered. 168 pediatricians completed the survey, giving a response rate of 52%. All of them had encountered a child with a presumed rare disease in their career, with 60% having seen one in the 6 months preceding the survey. The commonest presumed rare condition encountered was genetic/metabolic, and the most common difficulty (97%) encountered was lack of access to diagnostic testing. A third of respondents reported that rare diseases were taught in university and 60% felt unprepared to look after these children. Three quarter of respondents could not access to experts to advise them on management. Presumed rare diseases are commonly encountered by pediatricians in Tanzania, and there are challenges in diagnostic testing, gaps in training, lack of confidence in providing care and inability to access experts on rare disease management. To improve care of children with rare diseases, diagnostic testing should be made available, accessible and affordable. A review of medical training curricula should be done to incorporate rare disease education and skill development. Platforms and pathways to connect pediatricians with regional and global experts should be put in place to provide timely and appropriate care to children with rare diseases.

**Data availability statement:** The dataset used for this study has been provided as supplementary material.

**Funding:** Partial funding for this work was provided by the Ali Kimara Rare Disease Foundation. The funders had no role in study design, data collection and analysis, decision to publish, or preparation of the manuscript. No additional external funding was provided for this study.

**Competing interests:** Two of the authors: MN and MA are board members of the Ali Kimara Rare Disease Foundation which provided partial funding for the study. These interests do not alter the authors' adherence to PLOS Global Public Health policies on sharing data and materials.

## Introduction

Rare diseases are a group of medical conditions that affect a small number of people in the population. Definitions of rare diseases vary globally, and different criteria are used to describe these conditions. In the United States, a condition affecting less than 200,000 people is considered rare, while in Europe rare disease is one affecting less than 5 in 10,000 people [1].

Globally, there are between 5000–8000 rare diseases that have been identified affecting between 350 and 475 million people. The burden of these conditions is presumed to be higher in African countries, which have limited capacity to diagnose these patients, as well as a lack of treatment options and high cost of care, resulting in economic hardships, severe complications, and subsequent disabilities and mortality [2].

The diagnosis of these conditions depends largely on laboratory and genetic testing because of limited clinical experience with their manifestations, resulting in delayed diagnosis and treatment. Lack of treatment guidelines and the rarity of expert clinicians with experience managing these conditions further compounds the problem [3].

In developed countries, Paediatric sub-specialists, including geneticists, hematologists and endocrinologists, provide care to children with rare diseases. However, in low- and middle-income countries like Tanzania, primary care providers, including medical officers and pediatricians are on the front line of care for affected children [4]; responsible for initial diagnosis, management and timely co-ordination of referral to tertiary facilities. Therefore, they need to have adequate knowledge and skills for clinical care and to provide counselling, particularly for conditions whose therapy is not accessible or not yet developed.

However, global data suggest that pediatricians and other specialist doctors are not adequately trained to recognize, diagnose, or manage these conditions. A nationwide survey of pediatricians from Australia showed that only about 50% of them reported that rare diseases were adequately covered during their training, while about a quarter felt unprepared to care for these children [5]. Similarly, over 90% of physicians undertaking specialization in Poland perceived their knowledge on rare diseases as poor and less than 5% felt confident of taking care of these patients [6]. A study from Spain found that less than a third of physicians reported receiving training on rare diseases during their undergraduate and postgraduate years [7].

Very few studies gauging healthcare provider knowledge are available from the developing world. One survey from India reported that only a third of physicians had some knowledge of rare diseases while almost 80% reported they had no resources to diagnose and manage these conditions and their institution had no continuing education program in rare diseases [8].

Apart from lack of adequate knowledge, there are many challenges that pediatricians face in the diagnosis and management of rare diseases. Studies from high-income countries, including Australia, have identified diagnostic delays, lack of available treatments, and uncertainty of where to refer as common challenges [5]. Lack of specialists was highlighted as a challenge in in Malaysia by Shafie et al,

reporting only 13 rare disease specialists in the country [9]. Many of these conditions, especially in the perinatal period, present with signs and symptoms that may mimic other conditions hence may easily be misdiagnosed [10]. Various interventions have been proposed to improve access to knowledge and improve the diagnosis of rare diseases including specific education on "red flags" or pointers to increase attentiveness to rare diseases [11].

In recent years, patient advocacy groups have accelerated the global progress in diagnosing and managing rare conditions by bringing together governments, patients and scientific communities [3]. In Tanzania, the rare disease movement has been driven by a patient advocacy group called Ali Kimara Rare Disease Foundation (AKRDF) since 2016 [12]. This group brings together families affected by rare diseases and well-wishers from different sectors including media and sports who raise awareness on the plight of families. This has further been supported by the formation of the Tanzania Human Genetics Organization (THGO) which brings together stakeholders in the field of genetics including geneticists, lab scientists, pharma experts and doctors. THGO provides technical expertise, undertakes research and advises communities and government on policies and guidelines related to rare diseases [13].

There is currently no data from Tanzania on the burden of rare diseases, the types of conditions or the challenges faced in the management. This study aimed to close this gap by determining the knowledge, experiences and challenges faced by pediatricians in Tanzania in managing presumed rare diseases. This is the first step in highlighting the burden of the problem by utilizing the perspectives of pediatricians who are the specialists most likely to encounter children with rare diseases. The data from this study will help to identify specific needs for training to improve knowledge and skills and the challenges cited will prioritize areas for improvement in diagnosis and management.

## Materials and methods

**Ethics statement:** Ethical clearance was obtained from the Muhimbili University of Health and Allied Sciences (MUHAS) Institutional Review Board (ref: DA. 282/298/01.C/1895) and the permission to approach pediatricians was obtained from the professional society: Pediatric Association of Tanzania (PAT).

**Study design, setting and population:** This was a nationwide cross-sectional survey conducted among pediatricians working in Tanzania (Mainland and Zanzibar) who are members of Paediatric Association of Tanzania (PAT) and were included in the official PAT WhatsApp social media platform. PAT is the official professional association operating in United Republic of Tanzania, it had 325 members at the time of conducting this study (informal communication with PAT leadership), who were licensed and practicing at all levels of the health-care system from primary health care to tertiary referral facilities. The PAT WhatsApp platform is utilized for communication among members for social, academic and clinical issues. Members of the group are strictly pediatricians and are added soon after graduation. Over 90% of the pediatricians in the country are part of the group; those who are not are either non-practicing or choose to stay away from social media. The platform is also used for seeking consultation and discussion of cases managed by pediatricians who are in the group, discussions about patients are conducted with strict adherence to ethical principles of not sharing information that could reveal patients' identities. We used the WhatsApp group to disseminate this survey since it provided the widest reach to the target population.

**Data collection instruments:** Data was collected using a questionnaire (attached as supporting information) which was adapted from the instrument used in a similar study conducted in Australia [5]. This tool was found to be appropriate for this study because it provided comprehensive questions that aligned with the objectives of our research, making it a suitable foundation for adaptation. Questions were modified to fit the Tanzanian healthcare context; modifications mainly involved changes in geographical regions, regulatory authorities and access to relevant health care information. The questionnaire was pilot-tested on 4 senior paediatric residents (postgraduate trainees) to ensure the meaning, content and flow of the questions was clear. We selected senior paediatric residents because they were almost graduating and would have relevant experience in understanding concepts and questions related to rare diseases. The online link was shared with them, they reviewed it, filled out all the sections and gave written feedback on each section of the questionnaire.

Adjustments that were made based on their input included the flow and structure of the questions and additions to the list of available resources.

The modified questionnaire included sections on:

1. Pediatricians' demographics and practice which asked questions about years of experience, site of practice and sub-specialty training

2. Pediatricians' experience with presumed rare disease patients including types of presumed rare diseases, how many patients they had seen and how frequently they manage them

3. Challenges encountered by pediatricians in managing children with presumed rare diseases.

4. Pediatricians' education, training and their self-reported knowledge of rare diseases which asked about rare disease education during undergraduate and postgraduate training as well as workshops or seminars related to rare diseases

5. Likelihood of use of resources which included questions about their preferred resource with information about rare diseases

6. Pediatricians' perceptions and confidence regarding providing care to children with presumed rare diseases which asked about access to experts, knowing the pathway to channel a child with rare disease and their preparedness to provide care

**Data collection process:**

The questionnaire was shared as an online link circulated to members in the WhatsApp platform. To encourage participation in the survey, zonal PAT representatives were informed about the survey and requested to remind their members to accept the link and respond to the survey. All pediatricians on the group were eligible to take the survey as long as they were in active clinical practice, this was indicated at the beginning of the online questionnaire. Duplicate responses were avoided by adjusting the form settings to allow only 1 response.

**Data handling and analysis:** Data was collected between 7th January 2024 and 4th February 2024 and was auto-populated onto a Microsoft online Excel sheet. The data were checked for completeness and consistency and subsequently coded and analyzed using inbuilt formulae in Excel. Missing data in specific sections was noted and was omitted during analysis. Data were summarized into frequency distribution tables and charts that described the responses of participants regarding various aspects of rare diseases.

**Ethical considerations:** Ethical clearance was obtained from the Muhimbili University of Health and Allied Sciences (MUHAS) Institutional Review Board (ref: DA. 282/298/01.C/1895) and the permission to approach pediatricians was obtained from the professional society: Pediatric Association of Tanzania (PAT). The introductory section of the questionnaire provided information related to confidentiality, risks and benefits (Image provided as supporting information). Written consent was not obtained but was implied if the pediatrician filled out the online form and submitted it after reviewing the information provided.

Confidentiality was maintained by not collecting data on any direct identifiers like names or email addresses. Study data was kept confidential and accessible only to the investigating team by use of password protected computers.

## Results

A total of 168 pediatricians completed the survey giving a response rate of about 52%. There was representation from all zones in the country with the majority (63%) coming from the Eastern zone where the commercial capital city – Dar es Salaam is located. Fig 1 shows the distribution of respondents from different zones of the country which is in keeping with the distribution of pediatricians in these areas.

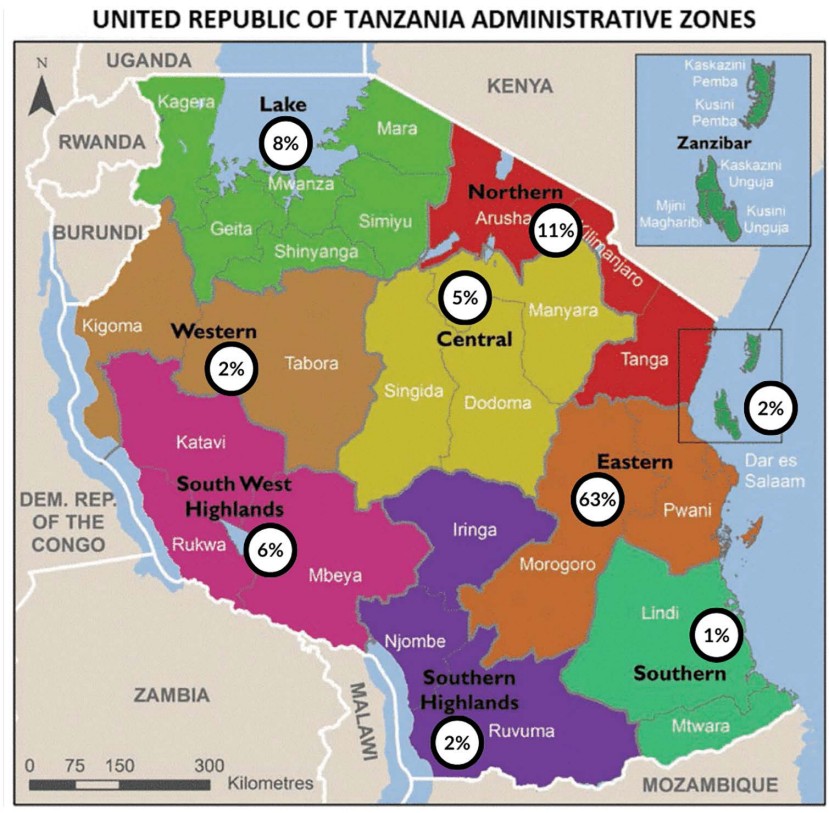

**Fig 1. Distribution of respondents from different zones in Tanzania.** The map illustrates percentage of respondents from the different geographical zones in Tanzania. Map modified from Mzee, 2021 [14] by adding percentages. Available from: https://doi.org/10.1186/s42269-021-00612-z. Available via cc by 4.0 license: http://creativecommons.org/licenses/by/4.0/.

Most participants (77%) were women and 79% were under the age of 45 years. Only about 30% had more than 10 years of clinical experience in the field of pediatrics. About half of the respondents worked in both public and private health care settings and the majority (81%) were general pediatricians with only a fifth being sub-specialized in a specific field of pediatrics. The characteristics of the participants are summarized in Table 1.

When asked how frequently rare diseases occur in Tanzania, most of the respondents (81%) believed that less than 5% of the Tanzanian child population is affected by rare diseases. Regarding their experiences in seeing children with rare diseases, all respondents had encountered a child with a presumed rare disease during their career but only 25 (15%) had seen more than 20 such children. Of these, about half of them were sub-specialists. Almost all (94%) reported looking after a child with an unusual cluster of symptoms and signs whose diagnosis could not be established and 22% reported having seen more than 10 such children. The most frequently encountered presumed rare conditions were metabolic/genetic (74%) followed by neurological, musculoskeletal and cardiac. Table 2 summarizes the experiences in encountering presumed rare diseases.

When asked about challenges encountered while managing children with presumed rare diseases, the commonest difficulty was lack of access to diagnostic and genetic tests with 97% of pediatricians encountering this difficulty. Other common challenges encountered included a lack of treatments (70%) and delay or inability to make a definitive diagnosis (66%). The challenges encountered are summarized in Fig 2.

**Table 1. Characteristics of study participants.**

| VARIABLE | N (%) |
|---|---|
| Gender | |
| Male | 39 (23%) |
| Female | 129 (77%) |
| Age groups | |
| <= 45 years | 132 (79%) |
| 46–60 years | 31 (18%) |
| >60 years | 5 (3%) |
| Years of experience as a pediatrician | |
| <5 years | 60 (36%) |
| 5–10 years | 52 (31%) |
| >10 years | 53 (31%) |
| Missing data | 3 (2%) |
| Type of practice | |
| Public sector only | 30 (18%) |
| Private sector only | 44 (26%) |
| Both public and private sector | 94 (56%) |
| Specialty | |
| General pediatrics | 136 (81%) |
| Pediatric Sub-specialty | 32 (19%) |

Regarding training received on rare diseases, only about a third of respondents said rare diseases were taught during their undergraduate curriculum and a similar proportion reported being taught during their postgraduate training. Only 30% of them had ever attended any workshops or lectures related to rare diseases.

On exploring their perceptions and confidence in recognizing and managing rare diseases, only about half of the respondents felt that their training was adequate for them to recognize patients with rare diseases and about 65% felt confident that they would know the referral pathway for a child with rare disease. About two-thirds (60%) felt unprepared to look after patients with rare diseases and over 70% did not have access to experts to support them in providing care. Responses to other sections of the questionnaire including confidence in finding rare disease information and perception on role of multidisciplinary care are further described in Fig 3.

When asked about the likelihood of using resources with information on rare diseases, over 80% of respondents favored a smartphone or tablet application while about 60% would use online modules via the Medical Council of Tanzania (MCT) website and a similar percentage would utilize face-to-face workshops or seminars. The likelihood of resource use by respondents are summarized in Fig 4.

## Discussion

This study was conducted to determine knowledge, experiences and challenges in managing presumed rare diseases among pediatricians in Tanzania. To the best of our knowledge, this is the first research study to investigate the experience of health care providers in managing rare diseases in Tanzania, the challenges faced in their treatment and the education and training about these conditions. There have been case reports and case series from Tanzania on individual rare diseases [15–17], however the overall spectrum of rare diseases has not been described. The findings of our study, having focused on pediatricians who are the specialists most likely to encounter rare conditions, can act as a proxy of the situation of rare disease knowledge, experiences and challenges in Tanzania.

**Table 2. Experiences of pediatricians in encountering presumed rare diseases.**

| VARIABLE | N (%) |
|---|---|
| Number of patients looked after with diagnosed rare diseases throughout career | |
| 1–5 patients | 88 (52%) |
| 6–10 patients | 38 (23%) |
| 11–20 patients | 17 (10%) |
| >20 patients | 25 (15%) |
| The number of patients looked after without definitive diagnosis reached | |
| 1–5 patients | 94 (56%) |
| 6–10 patients | 36 (22%) |
| 11–20 patients | 19 (11%) |
| >20 patients | 19 (11%) |
| Types of presumed rare conditions encountered[1] | |
| Metabolic/genetic | 124 (74%) |
| Neurological | 90 (54%) |
| Musculoskeletal | 59 (35%) |
| Cardiac | 59 (35%) |
| Endocrine | 55 (33%) |
| Immunological | 53 (32%) |
| Dermatological | 54 (32%) |
| Gastrointestinal | 47 (28%) |
| Respiratory | 46 (27%) |
| Cancer | 45 (27%) |
| Connective tissue | 42 (25%) |
| Renal | 38 (23%) |
| Rheumatic | 33 (20%) |
| Infectious | 26 (16%) |
| Hematologic | 2 (1%) |
| When the most recent suspected or diagnosed rare condition was seen | |
| <6 months ago | 101 (60%) |
| 6–12 months ago | 46 (27%) |
| 1–3 years ago | 12 (7%) |
| >3 years ago | 9 (6%) |

[1] Multiple selection was allowed for this question.

All participants in our study reported having managed presumed rare disease patients during their career; however, only 25% of participants had managed more than 10 children with presumed rare diseases. These findings reflect on the overall prevalence of these conditions; individually they may be rare, but as a group they are encountered in clinical practice although infrequently; similar to the findings by Rohani-Montez et al in a study conducted among clinicians in Europe and the United States where more than half reported that they have rarely (1–2 patients per year) seen patients with a rare disease [18]. However, the findings may also imply lack of index of suspicion for rare diseases hence the inability to recognize them.

It is plausible to attribute the reported experience of managing a higher number of children to sub-specialty training; 19% of participants had sub-specialty training while 25% of participants had managed more than 10 children with rare

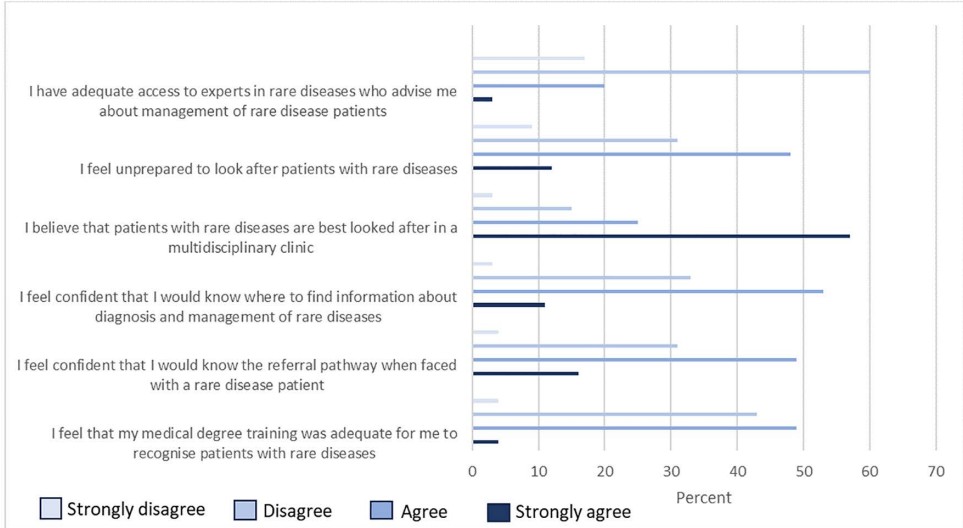

**Fig 2. Challenges encountered by pediatricians in managing presumed rare diseases.**

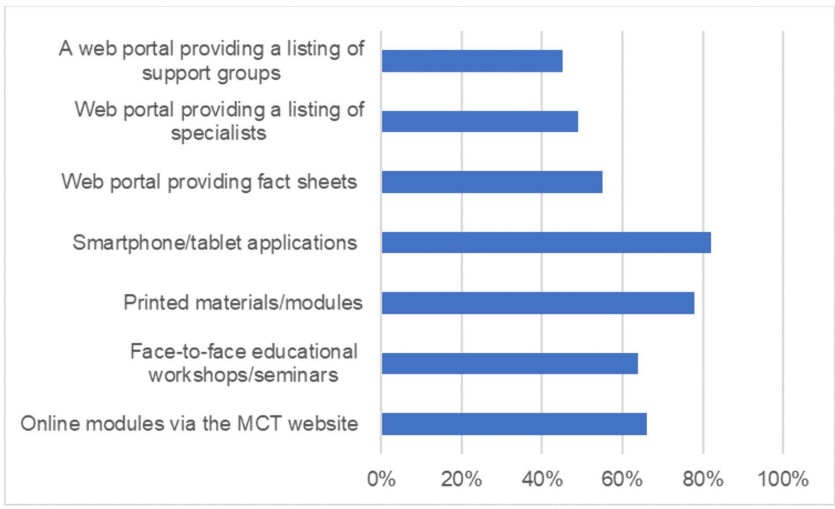

**Fig 3. Perception and confidence in caring for children with rare diseases.**

disease. Healthcare providers with sub-specialty and/or multiple specialties have better knowledge and experience in managing children with rare diseases and are usually practicing in referral and specialized healthcare facilities [19].

Limited access to diagnostic testing was the challenge most commonly faced by the participants in this study. This is because of lack of availability of genetic testing in Tanzania and the added high cost of testing if the sample is sent outside the country. Additionally, insurance companies frequently do not cover these costs. The difficulty in accessing genetic testing has been highlighted as a problem even in high income countries as reported by Zurynski in Australia [5]. Additionally, pediatricians in Australia also reported delays in interpretation of genetic tests; highlighting that apart from the physical diagnostic infrastructure, equally important is the human resource of clinical geneticists. In Tanzania, clinical genetics is a specialization that is not readily available with only 1 specialist in the country.

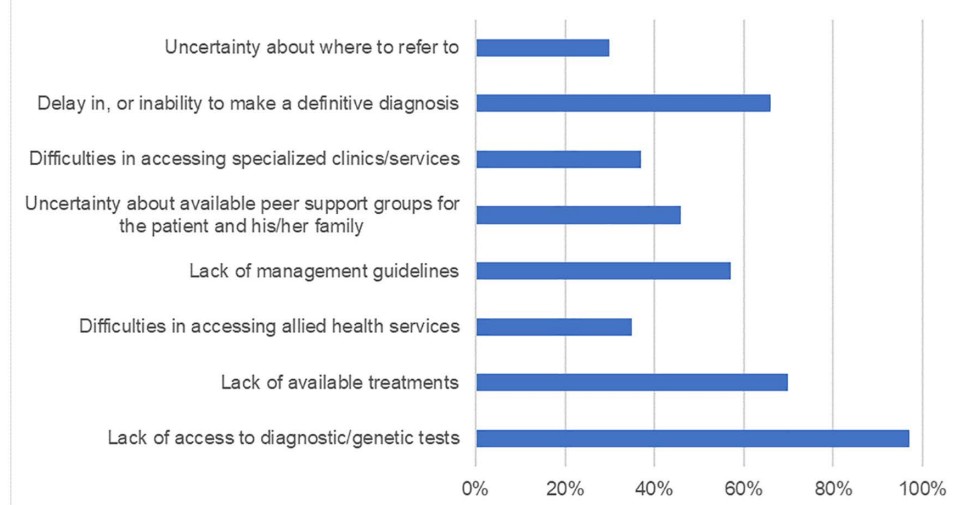

**Fig 4. Likelihood of resource utilization of pediatricians managing rare diseases.**

Comparative data from other low- and middle-income countries (LMICs) highlight similar challenges in the diagnosis and management of rare diseases in children. Studies conducted in LMICs indicate that pediatricians in these regions face significant barriers, including limited access to diagnostic tools, a lack of specialized expertise, and inadequate training on rare diseases—findings that closely align with our study in Tanzania. For instance, research from sub-Saharan Africa and Southeast Asia has noted the scarcity of genomic sequencing availability and the higher costs associated with diagnostic testing, contributing to delayed diagnosis and treatment [20].

Pediatricians in this study expressed a lack of training on rare diseases in both undergraduate and postgraduate training. There was also poor confidence in diagnosing and managing children with rare diseases with the majority indicating a lack of confidence. There is also lack of awareness of the global epidemiology with most pediatricians believing that less than 5% of Tanzanian children are affected by rare diseases; which is lower than the estimated global prevalence of rare diseases of between 6–8%. Similar findings have been documented from other studies globally, indicating low knowledge and ability to diagnose patients; this may consequently result in delays in diagnosis and treatment [21,22]. These gaps in training and subsequent confidence may be due to lack of updating of training curricula in keeping with progress in medical science especially genetics.

Capacity building for managing children with rare diseases in the form of guidelines, treatment, and care pathways with improvement in diagnostic facilities including treatment networks have been advocated for in several studies [23,24]. Participants in our study suggested various methods that can be used to improve their knowledge and skills on rare diseases. Use of a smart phone application with all relevant information on rare diseases is the resource most favored by pediatricians in Tanzania. This is likely due to the current age of digitalization and familiarity with technology. Majority of our study participants were less than 45 years of age and this is a generation which is comfortable and confident in using smart phone applications.

Other resource options include physical and virtual workshops, and availability of materials through web portals. Similar approaches were reported in an online survey which was conducted among clinicians in Europe and the United States who preferred a comprehensive online learning platform with current education on rare diseases [17]. It is therefore imperative for stakeholders involved in the care of these children including patient support groups, paediatric associations, medical schools, hospitals and the Ministry of Health to strategize and organize coordinated educational programs.

Most participants in this study agreed that children with rare disease should be managed by a multidisciplinary team, however less than a quarter of them had access to experts who could advise on the management of these patients. This shows a possible gap in coordinated care at different levels of the health system but could also be a reflection of lack of specialized pediatricians with the required knowledge and skills to assess and manage rare diseases. Globally, there has been a move to develop rare disease clinical networks that span across continents and harness the expertise of clinicians and researchers to provide the best possible care to patients [25].

Rare diseases cause significant burden to the patients, families, communities and health systems. Diagnostic delay is costly to the families and health systems and this is worsened by the poor knowledge and skills of practitioners [18,26,27]. These diseases contribute significantly to morbidity and mortality among children including infants and neonates [28], it is therefore important to consider strategies for mitigating existing challenges with a priority on capacity building for health-care providers.

Our study has some limitations to note: the response rate was 52% hence may not be reflective of all pediatricians in Tanzania. Additionally, given that a significant proportion (79%) of respondents were under 45 years of age, there is a possibility that the study may be biased toward younger pediatricians. Pediatricians with sub-specialty training were 19% and they are more likely to recognize and manage rare disease compare to general pediatricians hence this may bias the findings. The use of small sample to pretest the questionnaire may have overlooked flaws and may influence validity and finally, the use of an online survey via a WhatsApp group may have excluded some eligible participants who were not on the group, as well as those who were not familiar or comfortable with the use of technology.

To address these limitations, future research could aim for a more representative sample, incorporating strategies to enhance participation from older pediatricians. Additionally, further studies could explore whether digital tools are equally acceptable across different age groups. Despite these limitations, the study provides valuable insights into the challenges faced by pediatricians regarding rare diseases and highlights a significant gap in training and diagnostic resources.

## Conclusions and recommendations

Our findings suggest that pediatricians in Tanzania frequently encounter children with presumed rare diseases, with genetic/metabolic conditions being the most common from their perspective. However, a significant challenge lies in the limited access to diagnostic testing and lack of experts, which hinders effective management of these conditions. Furthermore, there is a notable gap in the training on rare diseases, leaving many pediatricians feeling unprepared to care for these children.

To improve the care of children with rare diseases in Tanzania, there is urgent need for national health care leaders and insurance partners to provide access to diagnostic testing and reduce the cost by providing subsidies and creating mechanisms through which these tests can be easily requested when needed.

Educational institutions should conduct a review of undergraduate and postgraduate medical curricula and incorporate rare diseases as an important component of training. Professional bodies within Tanzania and the wider African region should work together to provide access to relevant expertise and networks of care can be developed to connect pediatricians with national, regional and global experts.

We recommend that future studies should focus on developing a registry to document the true spectrum of rare diseases in Tanzania. Population based studies should be conducted to quantify the prevalence and burden of rare diseases and genetic and genomic research should be encouraged for accurate identification and risk stratification. Exploring the experiences of families of children with rare diseases is equally important to identify their challenges and needs for improved clinical care.

## Supporting information

**S1 Data. This is the complete dataset of the findings of the study.**
(CSV)

**S1 Text. This is the questionnaire that was administered via an online link to participants.**
(PDF)

**S1 Fig. This shows the introductory section of the online questionnaire indicating the information on benefits, risks and confidentiality.**
(TIF)

## Acknowledgments

The authors acknowledge the support from the Paediatric Association of Tanzania in conducting this study.

## Author contributions

**Conceptualization:** Mariam Noorani, Mohamedraza Ebrahim, Francis Furia.

**Data curation:** Mariam Noorani, Mohamedraza Ebrahim, Francis Furia.

**Formal analysis:** Mariam Noorani.

**Funding acquisition:** Mariam Noorani.

**Methodology:** Mariam Noorani, Mohamedraza Ebrahim, Francis Furia.

**Project administration:** Mariam Noorani, Francis Furia.

**Validation:** Mohamedraza Ebrahim, Francis Furia.

**Writing – original draft:** Mariam Noorani, Mohamedraza Ebrahim, Francis Furia.

**Writing – review & editing:** Mariam Noorani, Mohamedraza Ebrahim, Francis Furia.

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
