## [Decision Letter · Decision Letter 0]

23 Apr 2025

PGPH-D-25-00096

Rare diseases in children - knowledge, practices, and challenges faced by Pediatricians in Tanzania

Dear Dr. Noorani,

Thank you for submitting your manuscript to PLOS Global Public Health. After careful consideration, we feel that it has merit but does not fully meet PLOS Global Public Health’s publication criteria as it currently stands. Therefore, we invite you to submit a revised version of the manuscript that addresses the points raised during the review process.

We look forward to receiving your revised manuscript.

Kind regards,

Kathleen Bachynski, PhD, MPH

Academic Editor

Journal Requirements:

Additional Editor Comments (if provided):

Thank you very much for submitting your manuscript to PLOS Global Public Health. Better understanding challenges faced by pediatricians in Tanzania in caring for children with rare diseases is an important global public health issue. Reviewers #2 and #3 had recommendations for strengthening the manuscript, including clarifying some specific methodological details, including the complete questionnaire/survey instrument as an appendix, and expanding on the discussion section. Comparing your findings to any available data from other lower-middle income countries would be particularly valuable. I would also advise elaborating a bit more on the limitations section of the discussion. Notably, given that the response rate was 52%, in what specific ways might the findings not be reflective of all pediatricians in Tanzania? For example, the majority of the respondents (79%) were under the age of 45 years. Is this representative of the population of pediatricians in Tanzania, or was the study biased toward younger respondents? If so, that might have implications for some of the findings (such as the preference for a smartphone or tablet application).

Therefore, I invite the authors to revise and resubmit the manuscript.

Reviewers' comments:

Reviewer's Responses to Questions

Comments to the Author

1. Does this manuscript meet PLOS Global Public Health’s publication criteria? Is the manuscript technically sound, and do the data support the conclusions? The manuscript must describe methodologically and ethically rigorous research with conclusions that are appropriately drawn based on the data presented.

Reviewer #1: Yes

Reviewer #2: Yes

Reviewer #3: No

2. Has the statistical analysis been performed appropriately and rigorously?

Reviewer #1: Yes

Reviewer #2: Yes

Reviewer #3: No

3. Have the authors made all data underlying the findings in their manuscript fully available (please refer to the Data Availability Statement at the start of the manuscript PDF file)?

Reviewer #1: Yes

Reviewer #2: Yes

Reviewer #3: No

4. Is the manuscript presented in an intelligible fashion and written in standard English?

Reviewer #1: Yes

Reviewer #2: Yes

Reviewer #3: No

5. Review Comments to the Author

Reviewer #1: A very good and important study highlighted the unmet research area on rare disease as explained by author to be the first study in the country which will inform the public policy on the rare study. It highlighted how the disease is of importance especially to healthcare workers. The validated tools adapted from Malaysia help to validate the findings for public use. It is among the importance paper to inform the policy and proper government planning

Reviewer #2: Congratulations to the authors for their research work. Please see queries below

METHODOLOGY

1.What was the determinant that led to the selection of the Australian instrument for the survey? Was it culturally and contextually fit for purpose?

2.Was the survey tool pilot tested? If so, any adaptations made?

3.How were paediatricians contacted? By phone call? email message? whattsapp?

RESULTS

4. Might it be possible to have a map illustrating the zones and regions where your respondents came from?

5. Might it be possible to have table 3 in form of a graph for ease of understanding?

6.The same recommendation is made for table 4

DISCUSSION

7.Any data from other low-middle income countries in the same area of interest that you can compare your findings with?

8.Please highlight recommendations for policy and future research

Reviewer #3: This study presents the evidence on the diagnosis and treatment of Rare Diseases by paediatricians in some regions of Tanzania. It is developed through a cross-sectional, nationwide survey of this medical profession. Individual parameters of each professional are recorded, as well as characteristics of the place and sector of work, and paediatric subspecialty.

It is a study that attempts to provide information on the capacity to detect rare diseases, which are sometimes undiagnosed and difficult to treat, depending on the pediatricians’ level of training. Some solutions for the referral and treatment of patients are proposed. However, there are methodological shortcomings that are difficult to remedy at this stage. The imprecision of the results obtained is a considerable weakness for publication in an impact journal.

Finally, it is essential to have an expert review and editing of the English language throughout the manuscript.

Major and minor comments:

INTRODUCTION:

Page 2 lines 37, 38... The statement made is not in accordance with what is known as a ‘rare disease’, rather there is little clinical experience in its manifestations and it is necessary to rely especially on genetic/molecular studies to confirm the diagnosis and consequent treatment of the entity.

Page 2 lines 45, 46... Write the bibliographic reference for this statement.

Pag 3 lines 47-50... The wording becomes somewhat monotonous due to the frequency of the verb ‘to report’ in the incorporation of incidence data.

Pag 3 lines 63-65... Explain the meaning of ‘Red flags’.

MATERIALS AND METHODS:

Pag 5 line 100... Item 2 ‘Education and knowledge of rare diseases’ is not specified, this item of the questionnaire needs to be better described.

Pag 5 line 103... Item 5 ‘Use and availability of resources’ is not clear, what does it refer to specifically?

Page 5 line 104...To specify item 6 ‘Perceptions and practices in the management of rare diseases’.

Pag 5 lines 106-108, The term ‘paediatric’ is written instead of ‘pediatric’, correct it or decide on the same version of writing for the whole text.

Pag 5 lines 108-109, With reference to the questionnaire/survey, it would be necessary to have access to its complete visualisation as an annexed document, like the other 2 incorporated. I think it is a mistake not to present the reader with the complete questionnaire, thus also answering the questions raised above.

Some explanation - a sentence - on the statistical handling of the data and the criteria for its application would be missing.

This section of Material and Methods seems to be poorly worked out.

RESULTS:

One of the limitations of the study is the low participation of paediatricians, only 52% of the total sample. This may affect the analysis and conclusions of the study.

Page 6 line 121... Clarify the statement in the sentence that the 168 paediatricians who answered the questionnaire represent 52% of the study population.

Pag 7 line 131... How was this ‘<5% of the population in Tanzania is affected by rare diseases’ obtained, the bibliographic reference would be missing. Could it be from Zurynski Y et al 2017? In Tanzania there is still no official registry known for rare diseases.

Table 1 row 3... Sort the data correctly. Clarify the meaning of ‘paediatric specialist’, use appropriate terminology to avoid confusion.

Table 2 rows 1 and 2... Participants seem to give similar answers to both questions. Treatment in equal proportions between diagnosed and undiagnosed cases is insubstantial. If it is not diagnosed as a rare disease, it is difficult to assess the comparison and the result is also meaningless.

Table 2 row 3... Some disease should be specified for each system or apparatus exposed, even if only provisionally.

Pag 9 lines 140-144... These data should be better incorporated in a table.

Table 4 row 6 and 7... This option does not seem to be applicable in a context where it is difficult to confirm the diagnosis of a rare disease as in Tanzania.

Although table 3 provides qualitative data due to the subjective assessment of the professionals. Authors no clarify results obtained because of insufficient quantifiable data.

DISCUSSION:

In general, the wording has some repetitions of words, and could be changed to synonyms to obtain more fluent sentences.

Pag 12 lines 189-192... These statements are not accurate - 19% and 25% are not the same - and should be justified with more objective data.

Pag 13 lines 193-195... It is important to soften the sentences avoiding words with absolute meaning such as ‘lack’, better to write ‘reduced’, ‘inadequate’ or ‘limited’.

CONCLUSIONS:

Conclusions would have to be rewritten more accurately, few and set out in list form to make it easier for the reader to understand their conclusive importance.

BIBLIOGRAPHY:

Revise the entire bibliography to correct the title of the journals with the corresponding acronyms and page spacing.

Please consider to add this reference:

Mashala EI, Brunet-Llobet L, Lapitskaya A, Balsells-Mejía S, Mrina O, Miranda-Rius J. Bone disease and oromaxillofacial disorders: a cross- sectional study in a Tanzanian pediatric population. Orphanet J Rare Dis. 2025;20(1):77. Published 2025 Feb 17. doi:10.1186/s13023-025-03563-0

6. PLOS authors have the option to publish the peer review history of their article (what does this mean?). If published, this will include your full peer review and any attached files.

Do you want your identity to be public for this peer review? For information about this choice, including consent withdrawal, please see our Privacy Policy.

Reviewer #1: No

Reviewer #2: Yes: Dr. Angela Nyangore Migowa

Reviewer #3: No

---

## [Decision Letter · Decision Letter 1]

31 Oct 2025

PGPH-D-25-00096R1

Rare diseases in children - knowledge, practices, and challenges faced by Pediatricians in Tanzania

Dear Dr. Noorani,

Thank you for submitting your manuscript to PLOS Global Public Health. After careful consideration, we feel that it has merit but does not fully meet PLOS Global Public Health’s publication criteria as it currently stands. Therefore, we invite you to submit a revised version of the manuscript that addresses the points raised during the review process.

We look forward to receiving your revised manuscript.

Kind regards,

Miguel Reina Ortiz, M.D., M.S., M.P.H., M.P.T., Ph.D.

Academic Editor

Journal Requirements:

Additional Editor Comments (if provided):

Interesting study, important information to sure. There are major concerns. Please address the following:

0. GENERAL

1. Confirm whether the data and text presented in this manuscript have not previously been published. Cover letter states that findings build upon previous published research. Clarification of this would be appreciated.

2. Confirm whether this manuscript (or any portion thereof) is being considered for publication elsewhere.

3. Use American English spelling.

I. ABSTRACT

1. Rephrase first sentence for clarity. Suggested to rephrase to something like this "Diagnosing and treating rare diseases in children remain major challenges for pediatricians across the world."

2. The second sentence state that the research gap is to understand how often Tanzanian pediatricians are met with rare diseases, yet the next sentence focuses the paper into other issues such as knowledge, challenges in treatment, etc. Which is the gap and the focus of this paper?

3. Fourth sentence, suggest replacing "This" with "This study presents the results of a..."

4. How many total pediatricians are there in Tanzania? What proportion of those are in the WhatsApp group?

4.a. - for the manuscript, not the abstract, make sure to discuss selection bias - how do pediatricians get added to the WhatsApp group? Who decides, and under what criteria, who is admitted into the WhatsApp group? Is it possible that a non-pediatrician is in the WhatsApp group? How do the pediatricians in the WhatsApp group differ from pediatricians outside the group? Why use WhatsApp group and no other more commonly used participant recruitment? - Some of this is already addressed in lines 100-107 but not all, make sure that all of this is adequately addressed.

5. for the manuscript, not the abstract, how do you reconcile that such a high proportion (88%) of pediatricians have encountered rare diseases in their practice, when, by definition, rare diseases show extremely low frequency in either incidence or prevalence?

6. For the manuscript, discuss how a phone app with information will meet the need of lack of access to diagnostic tests.

7. Discuss the limitations of the study in the abstract.

II. INTRODUCTION.

1. The citation offered for the definition of rare diseases does not link to a European definition of rare diseases. Either add a citation for the European definition OR move the citation to the right location in the sentence and add a citation for the European definition in the right location of the sentence. Also, please cite either peer-reviewed literature OR a widely accepted textbook on rare diseases for definition, as opposed to a website. See, for instance, https://doi.org/10.1016/j.jval.2015.05.008

2. Citation 2 (Adachi et al) contains information that is either contradictory or different or lacking to what is stated in the text (esp. regarding number of rare diseases and percentage of global population affected by it). Please update the information in the sentence OR add appropriate citation (if new citation is added, ensure that numbering is updated AND that the new citation is either more recent or more rigorous on its methodology than Adachi et al).

3. Sentences in lines 37-40 are missing citation. Please add.

4. Citation 3 in line 44 does not match the text of the sentence. Ensure that all citations match the sentence correctly.

5. Line 44, specify what "our setting" mean.

6. Rephrase line 61 for clarity.

7. Line 62, comma after "Australia" is missing.

8. Lines 62-66, explain how the information from these studies reflect conditions in which adequate knowledge exist OR put the information in the right context.

9. Line 68, replace "red flags" for appropriate medical term.

10. Lines 73-74, text almost verbatim from Stoller - please rewrite this paragraph.

11. Please specify who are the members of THGO. Try to contrast membership, mandate and activities of AKRDF and THGO for the reader to understand the context of the field in Tanzania better.

12. Sentence ending in line 89 is missing citation.

13. As mentioned above, clarify research gap and study aim AND ensure consistency.

14. Line 95 - is it certain that this data WILL inform medical education? If not, then replace will by another appropriate modal verb or appropriate sentence.

III. MATERIALS AND METHODS

1. Add sub-headings. At minimum, Setting and Study Population, Sample Size and Sampling Methodology, Data Collection Instruments, Data Collection Process, Data Manipulation and Data Analysis, Ethical Considerations.

2. Line 98, delete "of" before "conducted."

3. Line 99. Suggest to place "mainland and Zanzibar" in parenthesis.

4. Line 101, add citation for the number of members in PAT.

5. Line 105. Review "who are the group," it does not seem to make sense in this sentence.

6. Line 105, do you mean "about" instead of "involving"?

7. Line 109. Replace "in the survey done in Australia" for a more formal writing style.

8. Line 111. Explain the "modification" process employed to adapt the survey to the Tanzanian context.

8.a. what frameworks or theories guided this process?

9. Line 112. Does the "questionnaire" refer to the original (i.e., Australian) OR the modified (i.e., Tanzanian) context?

10. Lines 113-119. Ensure consistency of use between "participant" and "pediatrician" - stick to one or the other.

11. Line 120. Replace "pilot tested" for "pilot-tested."

12. Line 120. Specify how many pediatricians participated in the pilot testing (and how were they selected).

12.a explain in detail the process, methodology and theoretical foundation of the pilot testing process.

13. What was learned from the pilot testing? Did some question need to be further adjusted? What input was received from participants in the pilot testing? Explain here.

14. Describe eligibility criteria.

14a. Were pilot testing participants allowed or not allowed to take part of the survey? Why or why not?

15. Line 125, replace "to" with "and" before "4th."

16. What software was used to collect survey data?

17. Line 126. How was incomplete data handled?

18. Line 131. Clarify what is meant by "implied" informed consent. Were participants asked for consent as part of the online survey? Show a screenshot of the survey's first page.

19. Describe data protection and cybersecurity steps taken to ensure privacy and confidentiality of data.

IV. RESULTS

1. Line 138. Text states "most regions" - clarify how many regions (out of how many are represented).

2. Figure 1 shows % from regions, how does this compare to the % of PAT members per region?

3. Line 139. The word "zones" is used whereas previously the word "regions" was used. Clarify and keep consistency.

4. Line 145. Describe which fields of pediatric subspecialty are represented.

4. Line 147 - Table 1. Add "missing" category for the "years of experience as a pediatrician" variable.

5. Line 148 - Table 1. Use formatting where only top and bottom lines for the table and for the headings is shown - eliminate lines in the middle (between rows and between columns) and the sides.

6. Line 150. Contrast participants' beliefs (re: % of Tanzanian population affected by rare diseases) with reality.

7. Line 153. Word "subspecialties" is used, whereas previously the word "specialties" (or derivatives) has been used. Recommend consistent use of "sub-specialties" (and derived words) across the document.

8. Line 155. Better explained what is meant by "the commonest rare conditions."

9. Line 158 - Table 2. Format as suggested above.

10. Line 158 - Table 2. "Number of patients ... without ... diagnosis" adds up to 99%. Please correct.

11. Line 158 - Table 2. "Types of ..." does not add up to 168 nor to 100. Explain with a footnote.

12. Line 158 - Table 2. "... most recent..." does not add to 100%. Please correct.

13. Line 160. Specify 97% of what?

14. Line 167. How many participants have attended CME or specialty/sub-specialty conferences where they could have been exposed to appropriate rare diseases content? Was this explored?

15. Line 168. Specify, half of what?

16. Line 170 and in general. Do not start sentences with numbers nor acronyms.

17. Line 171. There is a reference to "domains" but domains explored have not been previously introduced. Explain domains (and theories the work draws from) in the methods section.

18. Figure 3 cannot be understood as is. Add appropriate contextual notes, legends, better categories, etc. to understand what information is being presented.

19. Line 1740177. Does this refer to currently available resources (i.e., apps or online websites) or resources they wished they had but they do not have access to right now?

20. Figure 3 does not match context described in text.

21. Figure 4 does not match context described in text.

22. Need to better explain and connect survey to results. Need to add phrases like, "when asked about XXXX" participants showed "XXX"

22a. in the methods section describe the structure of the survey. For instance, Part 1 asked questions about XXXX such as "XXXXXX;" Part 2 asked XXXXXX, etc.

V. DISCUSSION.

1. Lines 180-181. Before the suggestion was made to clarify research gap and survey intent. Make sure that this line remains consistent with any changes made.

2. Lines 181-185. There is no need to restate the results, there is need to contextualize them (how do these compare to previous research? if there are differences, why would those differences exist?).

3. Line 186. This sentence is too broad (i.e., "the realm of rare diseases in Tanzania"), narrow it down.

4. Line 187, Where? in Tanzania?

5. Lines 186-196. The focus is being lost from the specifics of this survey to the broader topic of rare disease. This survey is not assessing the burden of rare diseases in Tanzania. Suggest to keep the discussion focused to the results of the study.

6. Lines 197-199, rephrase for clarity.

7. Line 200. This registry seems to be coming "out of the blue" - need to discuss it in intro and methods/results (as available) to give context to this sentence (add more context about the registry in the discussion as well).

8. Line 201. "1-3" patients is not one of the categories presented in the results. Explain this.

9. Line 202. Results presented here do not correspond to results presented previously. Please correct.

10. Lines 205-211. Make sure that these results are the same that were presented earlier in the paper.

11. Lines 216-218. How would this be in Tanzania?

12. Line 220. Delete "This was also mirrored in their responses."

13. Line 221. Specify "similar reports."

14. Line 226. Specify "similar interventions."

15. Line 231. Specify "similar approaches" in all these 3 last items compare results from this study with reports from the literature, specifically, what is similar, what is different and offer potential explanations for differences.

16. Line 237. Delete "this is an expected finding."

17. Lines 253-261. Move to the appropriate sections of the discussion where those issues are being compared to HIC.

18. Line 262. Remove "few." There are more than a few limitations.

19. discuss how using WhatsApp may have excluded people who don't have access to it OR do not feel comfortable using it OR are not "tech savvy" - use appropriate descriptor for the last.

20. Line 274, start with "our findings suggest" as opposed to "it is evident."

21. The conclusions is mostly repetitive of results. Rewrite to explain one solid conclusion and recommendation that can be derived from the results. It is not advisable to focus on a phone app as that was not the main focus of the study.

Reviewers' comments:

Reviewer's Responses to Questions

Comments to the Author

1. If the authors have adequately addressed your comments raised in a previous round of review and you feel that this manuscript is now acceptable for publication, you may indicate that here to bypass the “Comments to the Author” section, enter your conflict of interest statement in the “Confidential to Editor” section, and submit your "Accept" recommendation.

Reviewer #1: All comments have been addressed

Reviewer #2: All comments have been addressed

Reviewer #3: All comments have been addressed

2. Does this manuscript meet PLOS Global Public Health’s publication criteria? Is the manuscript technically sound, and do the data support the conclusions? The manuscript must describe methodologically and ethically rigorous research with conclusions that are appropriately drawn based on the data presented.

Reviewer #1: Yes

Reviewer #2: Yes

Reviewer #3: No

3. Has the statistical analysis been performed appropriately and rigorously?

Reviewer #1: Yes

Reviewer #2: Yes

Reviewer #3: No

4. Have the authors made all data underlying the findings in their manuscript fully available (please refer to the Data Availability Statement at the start of the manuscript PDF file)?

Reviewer #1: Yes

Reviewer #2: Yes

Reviewer #3: No

5. Is the manuscript presented in an intelligible fashion and written in standard English?

Reviewer #1: Yes

Reviewer #2: Yes

Reviewer #3: No

6. Review Comments to the Author

Reviewer #1: The author addressed all the comments provided and I recommend the paper to be pubished for widely dissemination as the findings are very important and have positive impact in policy implication to inform intervention in rare disease

Reviewer #2: (No Response)

Reviewer #3: Despite having given the authors the opportunity for a major revision, I regret to report that:

(i) Some of the responses to my comments have been really inconsistent. (ii) This paper has serious methodological weaknesses that cannot be remedied at this stage. (iii) The contribution is, quite frankly, poor and confusing for publication in an impact journal.

7. PLOS authors have the option to publish the peer review history of their article (what does this mean?). If published, this will include your full peer review and any attached files.

Do you want your identity to be public for this peer review? For information about this choice, including consent withdrawal, please see our Privacy Policy.

Reviewer #1: No

Reviewer #2: Yes: Angela Nyangore Migowa

Reviewer #3: No

Figure Resubmissions:

---

## [Decision Letter · Decision Letter 2]

22 Jan 2026

PGPH-D-25-00096R2

Rare diseases in children - knowledge, experiences and challenges faced by Pediatricians in Tanzania

Dear Dr. Noorani,

Thank you for submitting your manuscript to PLOS Global Public Health. After careful consideration, we feel that it has merit but does not fully meet PLOS Global Public Health’s publication criteria as it currently stands. Therefore, we invite you to submit a revised version of the manuscript that addresses the points raised during the review process.

We look forward to receiving your revised manuscript.

Kind regards,

Baldeep Kaur Dhaliwal, PhD

Academic Editor

Journal Requirements:

Additional Editor Comments (if provided):

N/A

Reviewers' comments:

Reviewer's Responses to Questions

Comments to the Author

1. If the authors have adequately addressed your comments raised in a previous round of review and you feel that this manuscript is now acceptable for publication, you may indicate that here to bypass the “Comments to the Author” section, enter your conflict of interest statement in the “Confidential to Editor” section, and submit your "Accept" recommendation.

Reviewer #1: All comments have been addressed

Reviewer #4: (No Response)

Reviewer #5: (No Response)

Reviewer #6: All comments have been addressed

2. Does this manuscript meet PLOS Global Public Health’s publication criteria? Is the manuscript technically sound, and do the data support the conclusions? The manuscript must describe methodologically and ethically rigorous research with conclusions that are appropriately drawn based on the data presented.

Reviewer #1: Yes

Reviewer #4: Partly

Reviewer #5: No

Reviewer #6: Yes

3. Has the statistical analysis been performed appropriately and rigorously?

Reviewer #1: Yes

Reviewer #4: No

Reviewer #5: No

Reviewer #6: Yes

4. Have the authors made all data underlying the findings in their manuscript fully available (please refer to the Data Availability Statement at the start of the manuscript PDF file)?

Reviewer #1: Yes

Reviewer #4: Yes

Reviewer #5: No

Reviewer #6: Yes

5. Is the manuscript presented in an intelligible fashion and written in standard English?

Reviewer #1: Yes

Reviewer #4: Yes

Reviewer #5: Yes

Reviewer #6: Yes

6. Review Comments to the Author

Reviewer #1: The Manuscript is getting in shape for publication as the author addressed all the comments given by reviewers

Reviewer #4: To my knowledge, this is the first national-level study from Tanzania examining rare diseases from the perspective of pediatricians, and it addresses a substantial evidence gap. The manuscript is ethically sound, transparently reported, and written in clear English. The dataset is openly available and complies with PLOS data-sharing requirements.

However, while the study is publishable in principle, it currently falls short of the analytical rigor expected for a Research Article in PLOS Global Public Health. The work is largely descriptive, and the available data are underutilized. This does not warrant rejection, but it does require major revision.

Reviewer #5: The manuscript tackles an important, under-studied topic, but clarity, rigor, and interpretability are limited. It offers valuable insights but needs substantial revision to strengthen structure, methodological transparency, data interpretation, and alignment of conclusions with findings.

Abstract: Unstructured; PLOS requires labeled sections (Background, Methods, Results, Conclusions). Background lacks clear justification for Tanzania and a defined knowledge gap. The Methods section in the abstract is missing. Key details such as study design, population, sampling approach, survey instrument, data collection period, and type of analysis are not described. Results include interpretive statements, and conclusions are broad with advocacy-style language. Minor grammatical errors and repetition reduce clarity. Minor grammatical and typographical errors (e.g., “60% of felt unprepared” knowledge, experiences and challenges … is not known”).

Introduction: Overly long with global context not fully linked to study aims. Rare diseases are not clearly defined, and the Tanzanian healthcare context is underexplored. The research gap is implied rather than explicit, and rationale for a survey approach is not justified. The introduction would benefit from greater conciseness, clearer definition of key concepts, stronger contextualization to Tanzania, and a more focused articulation of the study’s knowledge gap and rationale.

Methods: The Methods section provides a general overview of the study design and data collection process; however, several methodological details are missing or insufficiently developed, limiting reproducibility and assessment of rigor. Important methodological details related to study design, sampling, and eligibility criteria are inadequately described. Use of the PAT WhatsApp group may introduce selection bias. Questionnaire validation and pilot testing are limited and not fully described. Inclusion and exclusion criteria are not fully specified. Data collection process: Measures taken to prevent duplicate responses are not described. No data management plan (e.g., data security, access control) is described. Data analysis is restricted to descriptive statistics without subgroup analysis, and Excel use is not justified.

Results: Entirely descriptive and self-reported; multiple-response questions and denominators are unclear. Subgroup analyses are missing, and potential biases (selection, recall) are not addressed. Figures and tables require clearer captions and consistent reporting.

Discussion: Overly descriptive and repetitive. Critical interpretation is limited, causal statements from cross-sectional data are inappropriate, and the Tanzanian healthcare context is underexplored. Biases and methodological limitations are incompletely discussed. Capacity-building recommendations are not critically evaluated for feasibility. Pilot testing was conducted among senior pediatric residents rather than practicing pediatricians, which may limit its relevance; this limitation is not acknowledged.

Conclusion/Recommendations: Summarizes key findings but repeats Results/Discussion. Self-reported findings should be interpreted cautiously. Recommendations need prioritization, clearer stakeholder roles, and discussion of feasibility. Study limitations should be explicitly acknowledged, and patient/family perspectives could be better linked to clinical care implications.

Reviewer #6: Generally, the manuscript is well-written; however, the limitations do not address the small pre-testing sample, which may have overlooked questionnaire flaws and contributed to validity concerns.

7. PLOS authors have the option to publish the peer review history of their article (what does this mean?). If published, this will include your full peer review and any attached files.

Do you want your identity to be public for this peer review? For information about this choice, including consent withdrawal, please see our Privacy Policy.

Reviewer #1: No

Reviewer #4: No

Reviewer #5: No

Reviewer #6: No

Figure Resubmissions:

---

## [Decision Letter · Decision Letter 3]

28 Feb 2026

PGPH-D-25-00096R3

Rare diseases in children - knowledge, experiences and challenges faced by Pediatricians in Tanzania

Dear Dr. Noorani,

Thank you for submitting your manuscript to PLOS Global Public Health. After careful consideration, we feel that it has merit but does not fully meet PLOS Global Public Health’s publication criteria as it currently stands. Therefore, we invite you to submit a revised version of the manuscript that addresses the points raised during the review process.

The manuscript has been further evaluated by two reviewers, and their comments are available below.

Could you please carefully revise the manuscript to address all comments raised?

We look forward to receiving your revised manuscript.

Kind regards,

Ilse Bloom

Staff Editor

Journal Requirements:

Additional Editor Comments (if provided):

Reviewers' comments:

Reviewer's Responses to Questions

Comments to the Author

1. If the authors have adequately addressed your comments raised in a previous round of review and you feel that this manuscript is now acceptable for publication, you may indicate that here to bypass the “Comments to the Author” section, enter your conflict of interest statement in the “Confidential to Editor” section, and submit your "Accept" recommendation.

Reviewer #5: (No Response)

Reviewer #6: All comments have been addressed

2. Does this manuscript meet PLOS Global Public Health’s publication criteria? Is the manuscript technically sound, and do the data support the conclusions? The manuscript must describe methodologically and ethically rigorous research with conclusions that are appropriately drawn based on the data presented.

Reviewer #5: Partly

Reviewer #6: Yes

3. Has the statistical analysis been performed appropriately and rigorously?

Reviewer #5: Yes

Reviewer #6: Yes

4. Have the authors made all data underlying the findings in their manuscript fully available (please refer to the Data Availability Statement at the start of the manuscript PDF file)?

Reviewer #5: Yes

Reviewer #6: Yes

5. Is the manuscript presented in an intelligible fashion and written in standard English?

Reviewer #5: Yes

Reviewer #6: Yes

6. Review Comments to the Author

Reviewer #5: • Lines 16-19: Slightly awkward phrasing; could be “…WhatsApp group, where most pediatricians are registered.

• Line 91-92: Please provide a reference to support the statement about AKRDF’s role in Tanzania’s rare disease movement.

• Lines 102-106: The sentence is long and not clear; consider splitting it.

• The introduction is overly long (five pages) and would benefit from substantial condensation. The authors should streamline the section, remove redundancies, and focus on the most relevant background to improve clarity and readability. Make the introduction a 2-page

• Figures 2, 3, and 4 do not have titles. Please provide appropriate titles and captions for clarity."

• Figure 1 does not appear essential to the manuscript and could be removed to improve focus and conciseness."

Reviewer #6: (No Response)

7. PLOS authors have the option to publish the peer review history of their article (what does this mean?). If published, this will include your full peer review and any attached files.

Do you want your identity to be public for this peer review? For information about this choice, including consent withdrawal, please see our Privacy Policy.

Reviewer #5: No

Reviewer #6: No

 Figure Resubmissions:

---

## [Decision Letter · Decision Letter 4]

29 Mar 2026

PGPH-D-25-00096R4

Rare diseases in children - knowledge, experiences and challenges faced by Pediatricians in Tanzania

Dear Dr. Mariam

Thank you for submitting your manuscript to PLOS Global Public Health. After careful consideration, we feel that it has merit but does not fully meet PLOS Global Public Health’s publication criteria as it currently stands. Therefore, we invite you to submit a revised version of the manuscript that addresses the points raised during the review process.

I have reviewed the prior four reviewer rounds the paper went through and appreciate that most reviewer's comments are accepted. Reviewers mostly have recommended the paper for further publication. I do note however that by the time I was appointed as the handling editor, the paper has now been reviewed multiple times since submission in 2025 till date. The revised version is much improved. While the authors use a What' app group to reach clinicians, the survey instrument used provides fair bit of important contextual information but didn't seek to adequately identify diagnosis of disease which would have been a truer representation of the presumed rare disease burden in Tanzania. Rather the paper lists important, but many are known barriers in care with a substantial number of clinicians reporting lack of diagnostic services as the barrier.  This is a known challenge that the authors highlight early on in the introduction, and one would expect to exist regardless of setting. As the paper has already been peer reviewed, my comments below are addressable and ask the authors to consider providing a revised paper

I also couldn't make visible links between the AKRDF group mentioned early on the introduction (lines 77) to how the study was then conducted. Please explain how the groups reference relates to the study.

Please format figure 4 to provide better contrast in reading.

We look forward to receiving your revised manuscript.

Kind regards,

Danish Ahmad, MBBS,MSc,MNAMS,PhD,IP-FPH(UK),FRCP(Edin),FRCP(Lon)

Academic Editor

Journal Requirements:

Comments to the Author

1. If the authors have adequately addressed your comments raised in a previous round of review and you feel that this manuscript is now acceptable for publication, you may indicate that here to bypass the “Comments to the Author” section, enter your conflict of interest statement in the “Confidential to Editor” section, and submit your "Accept" recommendation.

Reviewer #5: All comments have been addressed

2. Does this manuscript meet PLOS Global Public Health’s publication criteria? Is the manuscript technically sound, and do the data support the conclusions? The manuscript must describe methodologically and ethically rigorous research with conclusions that are appropriately drawn based on the data presented.

Reviewer #5: Yes

3. Has the statistical analysis been performed appropriately and rigorously?

Reviewer #5: Yes

4. Have the authors made all data underlying the findings in their manuscript fully available (please refer to the Data Availability Statement at the start of the manuscript PDF file)?

Reviewer #5: Yes

5. Is the manuscript presented in an intelligible fashion and written in standard English?

Reviewer #5: Yes

6. Review Comments to the Author

Reviewer #5: The introduction has been substantially improved and is now clearer and more coherent. The authors should still review minor grammar and style issues, such as hyphenation in ‘low- and middle-income countries’ and sentence structure, to ensure consistency and readability.

7. PLOS authors have the option to publish the peer review history of their article (what does this mean?). If published, this will include your full peer review and any attached files.

Do you want your identity to be public for this peer review? For information about this choice, including consent withdrawal, please see our Privacy Policy.

Reviewer #5: No

Figure Resubmissions:

---

## [Editor Report · Decision Letter 5]

22 Apr 2026

Rare diseases in children - knowledge, experiences and challenges faced by Pediatricians in Tanzania

PGPH-D-25-00096R5

Dear Dr Mariam,

We are pleased to inform you that your manuscript 'Rare diseases in children - knowledge, experiences and challenges faced by Pediatricians in Tanzania' has been provisionally accepted for publication in PLOS Global Public Health.

Best regards,

Danish Ahmad, MBBS,MSc,MNAMS,PhD,IP-FPH(UK),FRCP(Edin),FRCP(Lon)

Academic Editor

The revised manuscript addresses my prior comments since handling editor & is acceptable for publication.

The publication editorial staff can confirm if the figure (1) titled ' Distribution of respondents from different zones in Tanzania' attribution has been adequately done as per journal's requirement and if there are any additional modifications required to the funding source.